# At the Cutting Edge against Cancer: A Perspective on Immunoproteasome and Immune Checkpoints Modulation as a Potential Therapeutic Intervention

**DOI:** 10.3390/cancers13194852

**Published:** 2021-09-28

**Authors:** Grazia R. Tundo, Diego Sbardella, Francesco Oddone, Anna A. Kudriaeva, Pedro M. Lacal, Alexey A. Belogurov, Grazia Graziani, Stefano Marini

**Affiliations:** 1IRCCS-Fondazione Bietti, 00198 Rome, Italy; diego.sbardella@fondazionebietti.it (D.S.); francesco.oddone@fondazionebietti.it (F.O.); 2Shemyakin-Ovchinnikov Institute of Bioorganic Chemistry of the Russian Academy of Sciences, Miklukho-Maklaya 16/10, 117997 Moscow, Russia; anna.kudriaeva@ibch.ru (A.A.K.);; 3Laboratory of Molecular Oncology, IDI-IRCCS, 00167 Rome, Italy; p.lacal@idi.it; 4Lomonosov Moscow State University, Leninskie Gory, 119991 Moscow, Russia; 5Department of Systems Medicine, University of Rome Tor Vergata, 00133 Rome, Italy; 6Department of Clinical Sciences and Translational Medicine, University of Rome Tor Vergata, 00133 Rome, Italy; stefano.marini@uniroma2.it

**Keywords:** immunoproteasome, ubiquitin–proteasome system, immune checkpoints, proteasome inhibitors, immunotherapy

## Abstract

**Simple Summary:**

Immunoproteasome plays a key role in the generation of antigenic peptides. Immune checkpoints therapy is a front-line treatment of advanced/metastatic tumors, and to improve its efficacy, a broader knowledge of the dynamics of antigen repertoire processing by cancer cells is mandatory. The scope of this review is to offer a picture of the role of immunoproteasome in antigen presentation to fuel the hypothesis of novel therapeutic interventions based on the modulation of this proteolytic complex and immune checkpoints.

**Abstract:**

Immunoproteasome is a noncanonical form of proteasome with enzymological properties optimized for the generation of antigenic peptides presented in complex with class I MHC molecules. This enzymatic property makes the modulation of its activity a promising area of research. Nevertheless, immunotherapy has emerged as a front-line treatment of advanced/metastatic tumors providing outstanding improvement of life expectancy, even though not all patients achieve a long-lasting clinical benefit. To enhance the efficacy of the currently available immunotherapies and enable the development of new strategies, a broader knowledge of the dynamics of antigen repertoire processing by cancer cells is needed. Therefore, a better understanding of the role of immunoproteasome in antigen processing and of the therapeutic implication of its modulation is mandatory. Studies on the potential crosstalk between proteasome modulators and immune checkpoint inhibitors could provide novel perspectives and an unexplored treatment option for a variety of cancers.

## 1. Introduction

Cancer immunotherapy is conceptually based on therapeutic stimulation of cancer immunosurveillance, a theory that states that tumor cells, mostly through the phenotypic alteration and the repertoire of tumor-associated neoantigens they often present, can be recognized and targeted by the immune system in the attempt to prevent disease progression. Nowadays, the dynamics of cancer and immune system crosstalk have been unequivocally unveiled to be extremely complex and has been renamed as cancer immunoediting. According to the three Es theory, this process comprises three defined phases: elimination of cancer cells by the immune system; equilibrium between tumor growth and control by the immune system; escape of neoplastic cells from immunosurveillance. The first two phases in some way coincide with the concepts described by the former theory, but the second phase can last long, running asymptomatically, until it is overrun by the third one, which results in tumor growth and dissemination. In an immunocompetent host, this breakthrough results from a positive selection of tumor clones that are able to evade the inhibitory control of the immune system by downregulating/masking antigen epitopes or by increasing the immunosuppressive properties of the tumor microenvironment [1,2].

Modern approaches of cancer immunotherapy, designed to restore a robust degree of immune activity against tumor cells, encompass immune checkpoint blockade, adoptive cellular therapies, and cancer vaccines [1,2,3,4,5]. Among these therapeutic interventions, immune checkpoint inhibitors (ICKi) have substantially revolutionized the oncology field by prolonging the survival of patients affected by highly aggressive/advanced stage cancers, such as metastatic melanoma and non-small-cell lung cancer (NSCLC). This approach is currently based on the use of monoclonal antibodies targeting inhibitory ICKs, such as CTLA4 (cytotoxic T lymphocyte antigen 4) and PD-1 (programmed cell death 1) or PD-L1 (programmed cell death 1 ligand) that regulate activated T lymphocytes function, switching off the immune response. A number of preclinical and clinical studies have revealed that antibodies raised against the immune checkpoint molecules enhance the antitumor immunity through not completely identified mechanisms of action, which differ depending on the target and specificity of the monoclonal antibody used (see Appendix A) [1,3,4,5]. Despite the undoubtful clinical success of ICK blockade, the on-field experimentation has opened several tasks that are worth being addressed: organ involvement and cohorts of signs and symptoms; second, the clinical efficacy is limited to subgroups of responder patients or, in many other subjects, after an initial response, drug resistance takes over. The latter event is generally due to the genetic and phenotypic heterogeneity of cancer cells and/or to tumor microenvironment remodeling during disease progression and dissemination [5,6,7]. Nonetheless, the overall efficacy of ICKi is affected by multiple factors, spanning from the expression and distribution of the target within the tumor microenvironment, the tumor mutational rate, and alterations of antigen presentation [8,9,10]. In this context, a crucial role is played by components of the host microenvironment that infiltrate the tumor and exert immunosuppressive effects, thus counteracting ICKi activity (i.e., infiltration by T regulatory cells (Tregs), dendritic cells, immunosuppressive myeloid cells, cancer-associated fibroblasts) [11].

Cancer cell immunogenicity is a key determinant for ICKi efficacy: malignancies which do not express tumor-specific antigens are not potentially susceptible to this approach. This is somewhat strengthened by the evidence that an immunotherapy-non-responsive cancer can turn into an immunotherapy-responsive one upon enhancement of tumor immunogenicity [8,9,10,12].

Hence, a better knowledge of the “immunopeptidome” (the repertoire of peptides bound to and presented by MHC molecules) and how tumor-specific antigen repertoire change during tumor progression is expected to improve the current therapeutic strategies; thus, a challenging issue is the identification and characterization of the MHC-I-presented peptides that modulate T cell-based tumor response [12,13,14].

At the molecular level, the generation of effective T cells that fight cancer requires a functional and efficient machinery for the multistep and multicellular mediated process, including antigen processing and presentation. The intracellular antigen processing pathway almost exclusively deals with the ubiquitin proteasome system (UPS) activity by which proteins are first ubiquitinated and then degraded. Indeed, the UPS carries out multiple functions in cells and in the onset and progression of different human pathologies spanning from neurodegeneration to cancer [15]. A major role in antigen processing is covered by a specialized and inducible form of proteasome (i.e., the multi-catalytic machine which degrades ubiquitinated proteins) named “immunoproteasome” [16]. This review focuses on immunoproteasome, its involvement in antigen generation, and on the therapeutic implications of its modulation to halt cancer progression. Finally, the potential crosstalk between proteasome modulators and immune checkpoint inhibitors is discussed.

## 2. Ubiquitin–Proteasome System: Cellular Biocomputing Machinery

In living cells, the proteome is constantly tuned through a complex, entwined, and multi-subcellular compartments network which coordinates the synthesis, folding, conformational upkeep and degradation of individual proteins [1]. The removal of undesired proteins is carried out by two main intracellular proteolytic systems, namely the ubiquitin–proteasome system (UPS) and autophagy [17,18,19,20]. The UPS is the major actor in the turnover of more than half of intracellular proteins that play fundamental roles in several facets of cell life, such as cell cycle, apoptosis, DNA repair, antigen presentation, inflammation, cellular response to environmental stress, and morphogenesis of neuronal networks [21,22,23]. The UPS’s hierarchical organization includes two intertwined and consecutive steps, i.e., the covalent ATP-dependent attachment of ubiquitin (Ub) polymers to a given substrate (target protein conjugation cascade), which is catalyzed by three classes of ubiquitin ligases, E1 (Ub-activating enzyme), E2 (Ub-conjugating enzyme), and E3 ligase, and its degradation by the 26S proteasome, followed by recycling of ubiquitin moieties along with the release of cleared protein oligopeptides (Figure 1) [15,24,25,26,27,28]. In the final step of ubiquitination, the E3 ligase, which is committed with substrate specificity [25,26,27,29,30] (Figure 1), mediates the formation of an isopeptide bond between the carboxyl C-terminal group of Ub and the ε-amino group of the lysine residue of the target protein. Thereafter, the reaction can be repeated multiple times, allowing the polyubiquitin chain to increase by 6 Ub moieties in average, in which each subsequent Ub monomer is connected to the previous one through an isopeptide covalent bond similar to that of the first Ub–substrate bond [30,31,32,33,34,35]. The process of ubiquitylation is a highly dynamic and reversible equilibrium; in fact, deubiquitinases or deubiquitinating enzymes (DUBs) can reverse the effect of E3 ligases by removing ubiquitin from target proteins. Furthermore, they mediate the polyubiquitin chain release during the hydrolysis of substrates by the proteasome [36,37].

Canonical 26S proteasome holoenzyme is a multifunctional proteolytic machine composed by the 20S proteasome core particle (CP), which features the proteolytic activity, capped by the 19S regulatory subunit (RP, also known as PA700), which carries out the ATP-dependent recognition, unfolding, and translocation into the 20S of the polyubiquitinated substrate [4,6,20]. The 20S core particle is a cylinder-like packed particle which contains four axial stacking heptameric rings arranged into two outer α rings and two inner β rings (i.e., α1–7β1–7α1–7β1–7) [38,39,40]. Eukaryotic 20S has a central channel, which houses six catalytically active β subunits, three for each β ring: the chymotryptic-like (β5 subunit, which hydrolyses at the C-terminus of hydrophobic residues), the trypsin-like (β2, which hydrolyses at the C-terminus of basic residues), and caspase-like (β1, which hydrolyses at the C-terminus of acidic residues) sites [40,41]. In the free 20S not engaged with a regulatory particle, the N-terminal tails of the α subunits point inwards to the center of the ring and the neighboring tails form an intricate network of inter-subunit interactions, constituting “the gate” of 20S, which regulates the substrate access through a 13 Å entry pore: the insertion of the substrate through this “N-terminal gate” is the rate-limiting step of proteasome activity [39,42,43]. Since “gate opening” is a key step in inducing the 20S function during evolution, cells have evolved different regulators (see also Section 2) which control this process and adapt proteasome functionality to the metabolic condition [44]. In fact, within the proteasome architecture, the outer α rings of the 20S make a nearly flat surface that binds the 19S and other alternative regulatory particles (i.e., PA28, see Section 3.2). The best known 20S activator is indeed the 19S which interacts, in the presence of ATP, with one or both ends of the 20S to form proteasome holocomplexes: the 26S, referred to as “single-capped”, and the 30S, named “doubly-capped” [44,45]. Once bound over the axial 20S pores, the 19S RP binding induces the displacement of the 20S α subunits’ N-terminal tails, thus opening the gate and promoting substrate access and translocation into the catalytic chamber [42,46,47]. From the structural point of view, the 19S is composed by two main elements, the lower “base” which directly binds to the 20S and the upper “lid”, thus forming a conformational dynamic complex [42,48]. The base consists of six structurally different subunits with ATPase activity (Rpt1–6), the structural subunits Rpn1 and Rpn2, and two ubiquitin-binding subunits, Rpn10 and Rpn13. The energy of ATP hydrolysis is spent to unfold the protein substrate and pull it down into the catalytic chamber of the 20S. Five ATPases (Rpt1–3; Rpt5 and Rpt6) present a hydrophobic tyrosine HbYX motif at the C-termini that insert into the α subunits pocket to induce gate opening.

Conversely, the peripheral lid consists of nine non-ATPase subunits: Rpn 3, 5–9, 11, 12, and 15, whose main functions are the strengthening of the 20S–19S interaction and deubiquitination of substrates before their processing by the ATPases [49,50,51]. Therefore, the 19S carries out different and fundamental functions: the recognition and unfolding of ubiquitinated substrates, the opening of the 20S pore, the entry of substrates into the catalytic chamber, and the release of ubiquitin moieties during substrate degradation [15,44,52]. Recently, the structure, function, and biogenesis of the 20S and 19S as well as the structural conformation of the proteasome holoenzyme have been extensively reviewed [6,20]. Emerging evidence shows that a protein tagged with at least four ubiquitin molecules is not the unique signal for proteasome degradation by 26S: in fact, multiple or single monoubiquitination appears to be sufficient to label a substrate for proteasomal degradation [53,54]. Additionally, a series of proteins has been reported to be degraded by the 26S regardless of ubiquitination [20,55,56], implying the existence of alternative molecular signals, such as a specific amino acidic sequence or structural elements (also called “degrons”), that mediate substrate recognition and degradation [19,56,57,58]. An additional topic that underlines the complexity of proteasome degradation pathway is the ubiquitin-independent degradation of macromolecular substrates by the uncapped 20S: several studies demonstrate that the 20S degrades natively unfolded and oxidative stress-damaged proteins [15,59,60,61]. The exact biological meaning and the molecular basis of these different degradation pathways are unclear and deserve additional investigations.

## 3. Hats off to Proteasome Variability

Since the UPS catalyzes the degradation of the majority of intracellular proteins and its organization is tuned on the cellular metabolic demands, the maintenance of an adequate activity is essential for cell homeostasis as much as an appropriate plasticity, in terms of structural and functional organization, is required for cells to adapt to the stimuli they experience [15]. Thus, the proteasome machine is a highly dynamic complex whose structural and conformational composition, substrate specificity is regulated at multiple steps encompassing transcriptional regulation, kinetics of assembly, post-synthetic modifications, and the interaction with a number of proteasome-interacting proteins (PIPs) which act as regulatory factors [40,62,63,64,65]. Focusing on the proteasome structural composition, two main elements of proteasome plasticity and variability are represented by (1) the interchangeability between constitutive and inducible catalytic subunits of the 20S; (2) the presence of different regulatory particles which can associate to just one or both free ends of the 20S. This allows generating different subtypes of proteasome that can coexist in a single cell and whose ratios may change among tissues. The metabolic and pathological stimuli that allow these canonical and noncanonical particles to form have been partially described, but to unequivocally address the interconnected, sometimes overlapping, or specific biological functions they carry out in vivo is a challenging task [62,66,67]. Noteworthily, in vertebrates, proteasome has gained considerable tissue specificity, as indicated by the existence of alternative forms of proteasome: immunoproteasome, also known as inducible 20S (i20S), thymoproteasome (Appendix B), and spermatoproteasome, in which the constitutive catalytic subunits of the 20S are replaced by inducible/tissue-specific homologs [62,68,69,70]. Immunoproteasome and thymoproteasome serve critical roles in the immunity, whereas spermatoproteasome is a testis-specific and chronologically-defined form of proteasome, exclusively identified in spermatocytes, spermatids, and sperm. It is characterized by the presence of a specific α4 subunit (α4s) (PMSA8 gene) that replaces the constitutive α4: the incorporation of this subunit into a newly formed 20S is mutually exclusive with wild-type α4 and seems not to alter the constitutive catalytic specificity of the 20S [69,70]. Nevertheless, α4s incorporation seems to promote the association of the 20S with an alternative regulatory particle named PA200, a nuclear-specific proteasome activator expressed in all mammalian tissues, but particularly abundant in the testis, where it plays a crucial role in spermatogenesis [71,72,73,74,75]. Remarkably, an increase of PSMA8 expression has been reported in different tumors, such as large B cell lymphoma, thymomas, and testicular germ cell tumors, even though its pathophysiological meaning and relevance as a novel therapeutic target have not been investigated yet [69,76,77]. As mentioned above, the reversible binding of activators (i.e., 19S and PA200) to either one or both α subunit rings of the 20S is another important level of proteasome organization that contributes to the overall heterogeneity of proteasomes (Figure 1) [15]. Besides the 19S, the best-characterized regulatory particle is PA28 (i.e., 11S regulatory particle), which preferentially binds the immunoproteasome, forming the PA28/i20S complex as discussed further (see Section 3.2) [16,78]. The binding of activators increases the proteolytic activity of the catalytic core, promoting the α-gate opening, influencing the substrate specificity of the complex, and, more importantly, affecting the repertoire of cleavage products [15]. Hybrid proteasomes (i.e., 19S–20S 11S, 19S–20S-PA200) have also been identified. Their biological function remains obscure, but their identification underlies how the cells edit the proteasome repertoire in relation to their specific needs [79,80,81].

### 3.1. Immunoproteasome as a Specialized Apparatus of Self-Target Designation

In the early nineties, proteasome was discovered as the crucial player for the class I MHC-restricted antigen processing pathway and two proteasome genes, namely *PSMB9* (*LMP2*) and *PSMB8* (*LMP7*), which encode two alternative subunits of the 20S, β1 and β5, respectively, were identified in close proximity of the transporter associated with the antigen-processing (*TAP*) gene in the MHC class II genomic region [82,83]. Concurrently, it was shown that synthesis and incorporation of these subunits into the 20S was driven by interferon γ (IFNγ) [84,85,86,87] (Figure 2). Thus, immunoproteasome, also known as inducible proteasome, is a specialized form of the 20S with a prominent role in immunity. Immunoproteasome preferentially and cooperatively incorporates three immune subunits, β1i, β2i (MECL-1), and β5i, to replace the constitutive catalytic subunits into the β ring of the 20S within its biogenesis pathway. The preferential assembly of inducible subunits is likely due to the higher affinity of β5i than of β5c for the proteasome maturation protein (POMP) which mediates the β ring formation [15,88,89]. The i20S assembly is four times faster than c20S, clearly reflecting the need for a rapid and transient response upon exposure to a proinflammatory stimulus. In fact, IFNγ induces, via the JAK/STAT signaling, the transcription of immune catalytic subunits, the *MHC-I* and *TAP* genes, thus enhancing the entire class I antigen presentation machinery [62,82,84,90]. Immunoproteasome is constitutively expressed at the basal level in hematopoietic cells and has a shorter half-life than c20S (average 27 h for immunoproteasome and 133 h for constitutive proteasome). Such a rapid turnover has the purpose of efficiently adapting to the environmental changes [91,92]. It has been shown that during the course of viral, bacterial, and fungal infections, immunoproteasome replaces up to 90% of the c20S pool [93,94]. As a matter of fact, besides the pioneering contribution of IFNγ, immunoproteasome was shown to be further transcriptionally induced by a plethora of inflammatory stimuli, such as IFNα and IFNβ, tumor necrosis factor α (TNF-α), lipopolysaccharides (LPS), as well as by redox unbalance [95,96,97,98]. It is important to recall that the different forms of proteasome particles can coexist inside the cells as well as the different peptide antigens generated. Anyway, in dependence to the different stimuli the cells are exposed to, the relative abundance of antigens generated by each specific subpopulation can be adapted [99].

A side-by-side comparison of the three different substrate-binding pockets of the c20S and i20S points out the enzymological differences of the two complexes [100]. In general, the i20S is characterized by increased chymotrypsin-like and trypsin-like activities, but a lower caspase activity [101]. In detail, the caspase-like subunit β1 accommodates peptides with an acidic residue in the P1 position, whereas β1i binds to peptides with a hydrophobic residue in the same position, exerting a branched-chain amino acid-preferring activity.

The folded trimeric complex formed by a given peptide and the MHC-I ligand cleft is exposed to the cell surface for presentation to the immune cells. The requisites for tight peptide MHC class I binding are essentially two: (1) the length of 8–9 amino acids and (2) an anchor of basic or hydrophobic residues located at the C-terminus or within the peptide sequence [68]. MHC-I does not accept C-termini with acidic anchor residues; thus, the substitution of β1c with β1i produces antigenic peptides with hydrophobic C-termini that can efficiently bind to MHC-1 molecules. Additionally, the structural properties of β5i also contribute to generating peptides with preferred C-terminal anchor amino acids for MHC-I molecules. In fact, the β5i S1 pocket accommodates larger hydrophobic amino acids chains than β5c (which presents, instead, a “small neutral amino acids-preferring activity”) and is characterized by a more hydrophilic environment around the catalytic threonine favoring the chymotryptic-like catalytic properties of the inducible subunit. Despite the β5 and β1 subunits, the active sites of β2c and β2i seem to be structurally identical, rendering this substitution more enigmatic, even though several studies reported an increase in trypsin-like activity of i20S with respect to c20S [68,100].

Additional forms of proteasome bearing a mix of standard and inducible subunits were identified. These intermediate proteasomes, which represent from one third to one half of the overall proteasome content in different tissues, such as liver, colon, and kidney, contain these triads of subunits, β1/β2/β5i or β1i/β2/β5i. Remarkably, a recently discovered mechanism of antigen generation through which proteasome increases the repertoire of antigens for presentation to the immune system is the “proteasome-catalyzed peptide splicing”: spliced peptides, which are made by two not contiguous fragments of parental proteins, are produced efficiently both by immunoproteasome and constitutive particles [102,103,104,105]. The existence of these mechanisms along with the copresence of different proteasome populations beyond the constitutive proteasome involved in antigen–peptide generation broadens the repertoire of antigens produced by a cell [92,106,107] (Figure 3). However, whether the incorporation of immune subunits triggers qualitative or quantitative effects on peptide repertoire generation is still not resolved since different studies report somewhat controversial results. In fact, a number of studies highlighted the positive role of immunoproteasome mainly against viral and bacterial antigens, whereas some studies reported that immunoproteasome expression can abrogate the presentation of some tumor epitopes [106,107,108,109,110,111,112,113].

Anyway, a defect in antigen presentation was found in the triple-inducible-subunit KO mice, and this alteration is now reported to be much broader qualitatively and quantitatively than that previously described in any of the β1i, β2i, or β5i single-subunit KO mice and still far greater than the sum of the defects these single-subunit KO animals were reported to bear [92,114,115]. Moreover, analysis of MHC class I-bound peptides shows that the antigen repertoire of KO mice differs from that of WT mice, reinforcing the hypothesis that immunoproteasomes generate peptides that, apparently, cannot be produced by constitutive proteasomes [108,116,117]. On the other hand, other studies suggest that immunoproteasomes affect the quantity rather than the quality of the given generated peptides, influencing also in this case the immune response [112,118,119]. Thus, some antigens are exclusively produced by the immunoproteasome or the constitutive proteasome, while others can be processed by both, and some others can be preferentially processed by intermediate-type proteasomes [120,121]. Of note, this distinction between the quantitative and qualitative effects of the antigen repertoire depending on the expression rate of immune–constitutive or mixed proteasome is not simply semantic. In fact, it is of basic significance not only to better understand the enzymatic properties of the different proteasome populations, but also to better define the MHC class I-dependent CD8+ T cell response in the context of specific physiopathological conditions [122].

As mentioned above, there is now extensive evidence that the UPS generates MHC class I-presented peptides; however, the source of the presented peptides is far less clear [123,124,125]. In this framework, the DRIP (defective ribosomal product) hypothesis is a novel concept about the source of peptides targeted by proteasome for antigen presentation. A considerable amount of newly synthesized polypeptides does not reach their native state after leaving the ribosome. It has been suggested that the defective proteins (indeed, DRiPs) are ubiquitinated and rapidly degraded by the proteasome and DRIPs-derived peptides may be preferentially loaded on class I MHC molecules through an unknown mechanism (possibly relying on the preferential uptake into the ER and/or exclusion of peptides derived from mature proteins) [126,127]. This proposed model challenges the older one in which MHC I-presented peptides are generated from the turnover of all cellular proteins. Therefore, further research is required to quantify the relative bulk of the newly synthesized and mature proteins in generating MHC class I-presented peptides by proteasome, and it will be important to unveil which proteins immunoproteasome shows cleavage preference for and how DRIPs-derived peptides produced by different proteasome populations are involved in mediating the immune response [124,127].

### 3.2. Regulatory Particles Which Associate with Immunoproteasome

As mentioned in the previous sections, besides the 19S, the second most common proteasome regulator is the ATP and ubiquitin-independent PA28 activator family that includes three highly homologous, ~28 kDa subunits, α, β, and γ, which form a ring-shaped 200 kDa multimeric complex (i.e., PA28αβ and PA28γ). This complex binds, in an ATP-dependent manner, to the two free ends of the 20S or associates with a single-capped 26S (19S:20S), forming the hybrid proteasome 19S–20S–PA28 [16,128,129,130,131]. This hybrid assembly hydrolyses tri- and tetra-peptides at a higher rate than constitutive 26S. The structure, function, and mechanism through which it induces the 20S gate opening was recently extensively and more competently reviewed [16]. PA28α and PA28β, which show a prevalent cytosolic localization, are upregulated by IFNγ and other proinflammatory cytokines. Consistently with this, their biological function is related to MHC-I antigen processing. Conversely, PA28γ is a primary nuclear homoheptamer, and its expression is not significantly upregulated by inflammatory stimuli. Thus, the role of PA28γ in antigenic processing is controversial, even though some studies have reported that alteration of PA28γ expression influences the bioavailability of epitopes derived from the pioneer translation products (PTPs)—polypeptides produced in the nucleus by a nonstandard translation process which occurs prior to mRNA splicing [78,120,132]. Interestingly, these epitopes show increased abundance in different cancers and are of key relevance in eliciting effective antitumor responses [133]. Remarkably, PA28γ binding to the i20S seems to mediate the ubiquitin-independent degradation of the key regulatory proteins and, thus, it is involved in different biological processes (besides the antigenic processing), spanning from cell growth and angiogenesis to apoptosis [79]. PA28αβ, a heteroheptamer assembled from four α (PSME1) and three β (PSME2) subunits, is constitutively expressed in lymphoid organs, but it can also be detected in tissues that lack immunoproteasome, where it likely exerts its biological functions also in association with the c20S. PA28αβ levels dramatically increase virtually in any other tissues in response to inflammatory stimuli, concomitant with the increase in other components of the MHC-I antigen processing pathway [120,134]. The i20S in association with PA28αβ mediates Ub-independent hydrolysis of the myelin basic protein during autoimmune neurodegeneration, accomplished through a novel class of charge-dependent proteasomal degrons [19,59].

Although several lines of evidence indicate the role of PA28αβ in MHC-I processing, its precise function and molecular mechanism of action are still poorly understood [78]. In 59fact, it has been reported that PA28αβ reduces the size and increases the hydrophilicity of peptides generated by i20S catalysis, thus producing peptides potentially suitable for MHC-I binding [62,129]. However, its genetic ablation does not lead to severe abnormalities in immune response against infections and causes the loss of only certain antigens. Nevertheless, this activator seems to stimulate the antigen production for some MHC-I alleles, but does not alter or downregulate the generation of other ones [78,110,135].

### 3.3. Immunoproteasome: Beyond Antigen Processing

In addition to the role in MHC-I antigen processing discussed above, immunoproteasome carries out other important functions in the dynamics of the immune system. In fact, its involvement in the molecular onset of autoimmune diseases has been proposed [136,137].

T cells knocked out for β2i, β5i, and, to a lesser extent, β1i show impaired proliferation and survival when transferred into virus-infected wild-type mice, suggesting a role in T cell expansion [138,139]. Moreover, ablation of the β5i activity (by either enzymatic inhibition or genetic depletion) suppresses the differentiation of Th-1 and Th-17 lineages and promotes the development of T regulatory cells (Tregs). The molecular basis of this intriguing β5i role remains to be determined: a major working hypothesis is that it may be involved in the turnover of factors that promotes Th1 and Th17 differentiation. This evidence about immunoproteasome involvement in T cells maturation and differentiation has fueled the hypothesis that the selective inhibition of β5i could be a suitable strategy in autoimmune disease therapy [114,116,140,141]. In this regard, a β5i selective inhibitor, ONX0914 (also called PR-957) has been shown to prevent the progression of rheumatoid arthritis in preclinical mouse models [141]. Moreover, β5i inhibition blocks the induction of experimental colitis, the development of lupus erythematosus-like disease, Hashimoto’s thyroiditis, and other autoimmune diseases [141,142,143,144,145,146]. Furthermore, the first selective inhibitor that targets all three proteolytically active immunoproteasome subunits (LU-005) has recently shown therapeutic efficacy against autoimmunity [124]. The UPS is well-known to cover a broad range of functions in immune cells of myeloid origin, being involved in the regulation of key crucial cell signaling pathways, such as NF-kB and IFN regulatory factors (IRFs), that mediate the production of inflammatory cytokines [15,113,147,148]. Interestingly, it has been reported that the natural inhibitor of NF-kB, IkBα, which is a prototypical proteasome substrate, is degraded by immunoproteasome faster than canonical proteasome and immunoproteasome asymmetrically capped with the 19S and PA28 (see also Section 3.2) [149,150,151,152,153]. Since immunoproteasome is constitutively expressed in myeloid cells, these results provide an intriguing hypothesis to explain the exceedingly rapid production of proinflammatory cytokines in response to various stimuli these cells are known to trigger [113,147]. However, the role of immunoproteasome in the myeloid maturation process is not completely understood. Despite this, the question whether immunoproteasome subunits affect NF-kB signal transduction differently from constitutive particles is still debated. In fact, studies concerning the effect of immunoproteasome subunits on NF-kB activation yielded contradictory results [154,155,156]. As a matter of fact, it has been recently reported that neither the kinetics of nuclear translocation nor the DNA-binding activity of NF-kB as well as the production of NF-kB-dependent proinflammatory cytokines differed between immunoproteasome-deficient (LMP2 KO and LMP7/MECL-1 double KO) and proficient cells [157].

Besides the role in the immune system, immunoproteasome seems to cover key functions in the regulation of protein homeostasis in the presence of redox unbalance, maintenance of stem cell pluripotency, neurodegenerative insults, muscle differentiation, and visual transmission [149,158,159,160,161]. Interestingly, concerning this last role, immunoproteasome is constitutively highly expressed in photoreceptors and synaptic regions of the immune-privileged retina, suggesting a role in normal neuronal functions of this highly differentiated nervous tissue [158,162]. Moreover, immunoproteasome is expressed in the lens and the cornea [163,164]. The precise role of immunoproteasome in eye functionality is still debated, even though it has been reported that its deficiency causes defects in bipolar response and abnormalities in retinal development [165,166]. Moreover, immunoproteasome expression is upregulated in stressed and injured retina, sustaining the investigation of the therapeutic potential of specific immunoproteasome inhibitors beyond the constitutive proteasome ones for the treatment of eye diseases [167].

## 4. Immunoproteasome: An Emerging Target in Cancer

Alterations of different genes belonging to the UPS are a hallmark of cancer. UPS dysregulation may occur at multiple levels, spanning from genetic modifications (i.e., mutations, amplifications, deletions), transcriptional network alterations (i.e., p53; NRF-1 and NRF-2) to epigenetic and post-translational modifications [15,168].

A number of studies underline the role of the i20S in cancer progression, strengthening its therapeutic potential. The tumor microenvironment is profoundly different from that of healthy tissues; this is also due to the presence of tumor-infiltrating lymphocytes (TILs) that release IFNγ and other inflammatory cytokines. Interestingly, immunoproteasome seems to possess both pro- and antitumorigenic properties, which are associated with the modulation of cytokine expression and tumor-associated peptide presentation, respectively [121].

Tumor cells can evade recognition by cytotoxic T lymphocytes (CTLs or CD8+ T cells) through downregulation of MHC-I at the cell surface or, additionally, by reducing immunoproteasome subunits expression [121,169] (Figure 4). In fact, non-small-cell lung cancer that undergoes the epithelial–mesenchymal transition shows a reduced immunoproteasome subunits expression: this leads to a dramatic drop of heterogeneity of the antigen/peptide repertoire produced by tumor cells and to poor clinical outcomes [170]. Thus, it has been proposed that a decrease in immunoproteasome expression might represent a mechanism of immune escape in tumor cells which present with a mesenchymal phenotype since this downregulation is associated with a decline in the amount and diversity of MHC-I-presented peptides [170]. In accordance with these data, transforming growth factor β (TGF-β)-induced epithelial–mesenchymal transition leads to a decrease in the immunoproteasome content [170]. Moreover, in the early stage of NSCLC, low expression of the i20S is linked to an increased risk of recurrence and metastasis onset [170]. At the molecular level, one proposed mechanism which links carcinogenesis with immunoproteasome deficiency is the differential expression of the β5i subunit. Two main β5i variants have been described, which are both induced by IFNγ: LMP7E2 and LMP7E1. LMP7E2, usually expressed in normal cells and in certain cancer types, is regularly incorporated into the mature i20S. However, many cancer cell lines express only the LMP7E1 isoform that does not interact with the 20S assembly chaperone POMP and thus cannot be integrated into the mature i20S, leading to a deficiency of functional immunoproteasome [171]. Moreover, a polymorphism at amino acid 49 of LMP7 (K49 instead of Q49), localized at the pre-sequence of β5i, reduces the rate of proteasome assembly and is associated with a higher risk of developing colon carcinoma [154]. On the other hand, overexpression of immunoproteasome subunits due to an increase in IFNγ production by TILs correlates with a better prognosis in different tumors, such as melanoma and breast cancer [121,168].

Immunoproteasome expression is not only triggered by paracrine production of proinflammatory cytokines (such as IFNγ) by immune cells, but it is constitutively elevated in hematological malignancies [155,172]. In myeloid leukemia cells, the i20S increase was associated with a higher survival rate [156]. Interestingly, the upregulation of immunoproteasome by IFNγ overcomes resistance to the proteasome inhibitor bortezomib and sensitizes hematological malignant cells (such as multiple myeloma and leukemia) to a selective immunoproteasome inhibitor ONX0914 [173]. This opens up the perspective of developing therapeutic approaches based on selective inhibition of immunoproteasome subunits different from those targeting the constitutive ones. Moreover, some evidence indicates that immunoproteasome alterations can have an impact on the onset of inflammation-driven carcinogenesis: indeed, β5i inhibition prevents colitis associated with colon carcinoma [144]. Thus, the emerging complex picture indicates that the altered expression of immunoproteasome subunits (mainly LMP2 and LMP7) is common in various tumors, but the extent of the expression and its biological significance vary depending on cancer type and grading [94,144]. As a matter of fact, the immunopeptidome changes in the context of tumor microenvironment and depending on the relative abundance of constitutive or inducible proteasome. For example, a number of cancer antigens derived from members of the melanoma antigen gene protein family (MAGE), whose expression is restricted to reproductive tissues but which are also aberrantly expressed in a wide variety of cancer types, such as MAGA3_(114–122)_, MAGEC_(42–50)_, and MAGEA2_(338–344)_, are produced by immunoproteasome but not by the constitutive proteasome [103,174]. Since the identification and characterization of neoantigens is of clinical relevance, modern strategies which combine genomic, proteomic, and immunopeptidomic approaches are a powerful way of discovering novel presented antigens and tumor-associated antigens, paving the way to the novel therapeutic potential [175,176]. As mentioned in the previous section, intermediate proteasomes broaden the repertoire of MHC-I antigenic peptides and, intriguingly, are involved in the production of unique tumor antigens (Figure 2 and Figure 3). In fact, it has been reported that some peptides derived from proteins belonging to the melanoma antigen gene (MAGE) family are generated by intermediate forms. Specifically, the β1i–β2–β5i intermediate produces the MAGE-A10_(254–262)_ peptide, whereas the β1–β2–β5i intermediates generate the MAGE-C2_(191–200)_ and MAGE-A3_(271–279)_ peptides. On the other hand, other antigenic peptides, such as MAGE-A3_(114–122)_ and MAGE-C2_(42–50)_, are produced with equal efficiency by the i20S and intermediate proteasomes [92,177,178]. Moreover, intermediate proteasomes were detected in a number of tumor cells, including lung carcinoma, myeloma, osteosarcoma, and melanoma [66,79,92,177]. Despite this evidence, the role of these forms of proteasome in cancer onset and development is poorly known.

### Immunoproteasome Inhibitors

The discovery and application in clinical practice of proteasome inhibitors have revolutionized the therapeutic approach of multiple myeloma, further improving the survival rate of patients with refractory or relapsed forms of this devastating tumor. Currently, three clinically approved proteasome inhibitors are available, namely (i) bortezomib (approved in 2003 and 2004 by the FDA and the EMA, respectively), (ii) carfilzomib (approved in 2012 and 2015 by the FDA and the EMA, respectively), and (iii) the first oral inhibitor, ixazomib (approved in 2015 and 2016 by the FDA and the EMA, respectively). As extensively reviewed elsewhere [15], proteasome inhibition results in multiple deleterious downstream effects in cancer cells, including downregulation of NF-κB signaling, alteration of cytokine secretion, stabilization of p53, and cell cycle arrest. A poorly explored issue that should be clarified concerns the contribution of proteasome inhibition to PD-L1 degradation and its impact on cancer immunotherapy [179]. PD-L1 undergoes ubiquitination and proteasome degradation by different E3 ligases (STUB1, Cullin3, and β-TrCP) [180,181,182]. Cancer cells exhibit the ability to inhibit this process with a consequent impairment of T cell activity. Remarkably, within the tumor microenvironment, TNF-α increase activates NF-kB in cancer cells, leading to the further increase of deubiquitinase CSN5 (COP9 signalosome 5) expression which inhibits PD-L1 degradation, facilitating immune escape of cancer cells. How proteasome inhibitors used in clinical practice impact PD-L1 degradation and, therefore, immune escape of cancer cells mediated by the PD-1/PD-L1 pathway is unclear and represents an important aspect to be investigated.

Despite the beneficial effects, the therapeutic potential of PI is limited by several drawbacks, including the low potency and specificity, the onset of adverse effects, and development of drug resistance [15,183]. Moreover, the main therapeutic benefits are limited to hematological malignancies since proteasome inhibitors are largely ineffective against solid tumors [184,185]. Therefore, there is a growing demand for novel inhibitors with different mechanisms of action and more favorable pharmacological profiles. To achieve this goal, one of the currently explored strategies is the development of selective immunoproteasome inhibitors. In fact, by targeting immunoproteasome subunits, it is expected to achieve a more selective inhibition of proteasomal activity in cancer cells, thereby widening the therapeutic window. Since immunoproteasome plays pivotal roles in antigen presentation, participates in a variety of immune processes and, in general, regulates protein homeostasis, selective inhibitors are expected to bring new therapeutic opportunities for the treatment of various diseases (beyond cancer), spanning from neurodegeneration to inflammatory and autoimmune disorders [186,187]. Therefore, several covalent and noncovalent peptidyl inhibitors selective for immunoproteasome subunits with different structural and biochemical properties have been developed and are currently studied. A number of detailed reviews on immunoproteasome inhibitors are already available [186,188,189,190,191]. Of note, ONX0914 was the first developed selective epoxy ketone peptide which showed an improved activity towards the β5i subunit (IC_50_ = 5.7 nM) with respect to β5c (IC_50_ = 54 nM). Starting from the success of ONX0914, a series of β5i-selective inhibitors with comparable activities and increased selectivity have been synthesized, such as PR-924, LU-015i, UK-101, LU-002i, and YU-102 [186]. Importantly, on the ONX0914 backbone, the only immunoproteasome inhibitor synthesized so far is KZR-616 that has entered the stage of clinical trials. In fact, several phase 1/2 trials are ongoing with the aim of evaluating the safety, tolerability, efficacy, pharmacokinetics, and pharmacodynamics of KZR-616 treatment in patients with autoimmune diseases (such as active polymyositis, dermatomyositis, lupus erythematosus) (NCT03393013, NCT04628936, NCT04033926) [186,192]. As already mentioned, a number of preclinical studies evaluated the potential of selective immunoproteasome inhibitors as anti-cancer therapeutic agents. It is important to note that the currently approved proteasome inhibitors target both constitutive proteasome and immunoproteasome subunits and this lack of selectivity, and primarily the inhibition of the constitutive proteasome, is postulated to account for the onset of several severe adverse events [189,193,194]. In fact, the target tissues of proteasome inhibitor toxicities predominantly express constitutive proteasomes; thus, selective inhibition of immunoproteasome might reduce or eliminate most of these toxicities [195,196]. Among the plethora of proposed immunoproteasome inhibitors for cancer treatment, the most promising ones, at least in preclinical models, are PR-924, which specifically inhibits the β5i subunit, and UK-101, that targets β1i. Importantly, both inhibitors reduce the proliferation rate of different tumor types (e.g., leukemia, multiple myeloma, and prostate cancer), including those resistant to bortezomib. Moreover, in mouse models, they are characterized by lower toxicities as compared to other therapeutic approaches and are generally better tolerated [186,197]. Interestingly, it has been demonstrated that exposure of multiple myeloma cells to bortezomib before administration of the immunoproteasome inhibitor ONX914 renders cells more sensitive to i20S inhibition, providing the biological rationale for using a combination therapy that includes selective immunoproteasome inhibitors and pan-proteasome inhibitors [198,199]. More recently, a novel, highly selective i20S inhibitor, M3258, was developed, characterized by a 500-fold greater specificity for the β5i subunit as compared to other proteasome inhibitors and oral bioavailability. In a xenograft model of multiple myeloma, M3258 showed strong target inhibition and in vivo efficacy, with some animals revealing complete tumor eradication. It also demonstrated an attractive overall profile with regard to physicochemical properties, biotransformation, and pharmacokinetics. Daily M3258 administration was associated with a more durable in vivo efficacy in multiple myeloma models compared with intermittent schedule, supporting the hypothesis of the need of continuous LMP7 inhibition to sustain tumor cell apoptosis. Additionally, lower systemic toxicity was observed than for other inhibitors of constitutive proteasome or immunoproteasome [200,201,202]. The robust efficacy of the M3258 inhibitor in preclinical models supported the rationale for a phase 1 clinical trial (https://clinicaltrials.gov/ct2/show/NCT04075721, accessed date: 25 July 2021) to determine the safety, tolerability, pharmacokinetics, pharmacodynamics, and early efficacy signs of this drug as a single agent (dose escalation) and as coadministered with dexamethasone (dose expansion) in patients with relapsed/refractory multiple myeloma [202]. Unfortunately, this trial was discontinued due to the changed therapeutic landscape and the lack of recruitment (see the ClinicalTrials.gov identifier accessed date: 25 July 2021). The overall bulk of described data indicate that the development and introduction in clinical practice of an immunoproteasome-specific inhibitor is largely awaited [189,203,204]. However, the onset of side effects and the extent of off-targets as well as the development of resistance following administration of selective immunoproteasome inhibitors remain unknown [200].

## 5. Immunoproteasome and Immune Checkpoint Inhibitors: A Glance to the Future?

As mentioned in the previous sections, the production of tumor-associated antigenic peptides recognized by CTLs is a process that starts in the cytoplasm with the degradation of cellular proteins mainly by immunoproteasome (see also Appendix B) [205,206]. Peptide antigens produced by cancer cells are commonly classified into two main groups, namely with high and low specificity. Antigens with high tumor specificity are encoded by viral genes (expressed only in infected cells), mutated genes (generated by the intrinsic instability of cancer cells and hereafter referred to as neoantigens), and cancer germinal genes (expressed as a result of genome-wide demethylation occurring in germinal cells) [99,207]. Moreover, it is known that tumorigenesis is strictly related to genetic diversity and high mutational burden of cancer cells, which increase the possibility of production of neoantigens [168,176,208,209]. This high mutational heterogeneity and neoantigens frequency positively correlate with the response to ICKi therapy. In fact, ICKi are particularly effective against cancers that present with a high burden of mutations and are characterized by DNA mismatch repair deficiency, such as colorectal cancer and NSCLC [210,211,212]. Thus, neoantigens have been proposed to be a prognostic marker for positive clinical outcomes [168,176,208,209]. As a matter of fact, one of the main reasons of acquired resistance to the ICKi therapy seems to be the loss of neoantigens recognized by circulating T cells, suggesting that tumors are “able” to eliminate some mutations during the acquisition of a resistant phenotype [212]. Remarkably, despite the approval of the ICK therapy for cancers characterized by a high mutational burden, a very recent study failed to support the concept that a high mutational burden is a positive biomarker for the ICKi treatment in all solid tumors. In fact, a high mutational burden seems to behave as a predictive marker of response to ICK-based therapies only when the CD8^+^ infiltration level correlates with the neoantigen load (such as melanoma, lung, bladder cancers, and colon cancer). On the other hand, for tumors in which no relationship between CD8^+^ levels and the neoantigens load is reported (such as glioma), the high mutational burden failed to predict a positive response to therapy [213]. Thus, it clearly emerges that additional tumor type-specific studies should be performed to unveil the role of this biomarker in the ICKi response. Anyway, the identification of an additional biomarker as well as of non-invasive techniques that monitor the microenvironment before and during the course of the treatment (e.g., imaging-based radiogenomics) are urgently needed for selecting patients who will benefit from immunotherapy [214,215].

In light of the plethora of antigenic peptides produced by the proteasome pathway, it is not surprising that alterations of the proteasome activity and composition could be linked to antigen processing and the ICKi response. Of note, the local production of IFNγ within the tumor microenvironment by infiltrating T lymphocytes positively correlates with clinical outcomes to the ICKi therapy and cancer vaccination in tumors like metastatic melanoma [216,217]. In fact, the upregulation and secretion of chemotactic cytokines (such as IFNγ and TNF-α) increase the recruitment of additional immune cells and alter the tumor microenvironment, stimulating the inhibition of immune exclusion of cancer signatures [218]. Interestingly, some studies show that the primary response to anti-CTLA4 antibodies required a high-level exposure of MHC-1 on the surface of cancer cells at baseline; on the other hand, the response to anti-PD-1 antibodies is linked to a pre-existing IFNγ transcriptome signature [219]. In a recent study, the transcriptome of baseline and on-therapy tumor biopsies from a cohort of 101 patients with advanced melanoma included within the CheckMate 038 study (https://clinicaltrials.gov/ct2/show/NCT01621490, accessed date: 25 July 2021) and treated with nivolumab alone or in combination with ipilimumab (see also Appendix A) has been analyzed [220]. These data, together with in vitro studies, suggest that the immune activation, which follows the administration of ICKi, is associated with the expression of IFNγ response genes, mediated by the increase of T cell infiltration. Among the different sets of genes induced by IFNγ, the most relevant in mediating the response to the therapy are those involved in antigen-presenting machinery [220]. Thus, a combination therapy of ICKi with agents that independently trigger the intratumoral production of IFNγ could become meaningful [220,221,222,223]. Consistently with the role of IFNγ signature in driving ICKi response, it has been proposed that upregulation of immunoproteasome subunits in tumor cells might be also involved in this process [216,220]. In fact, the local production of IFNγ induced by ICKi, which are routinely used in the treatment of advanced/metastatic melanoma, leads to the modulation of proteasome composition, thus inducing the generation of antigenic peptides [216,217]. Consistent with this observation, the expression of immunoproteasome subunits β1i and β5i was associated with a better prognosis in the case of tumors with a high mutational burden (i.e., melanoma and NSCLC) and positively correlated with the response to ICKi and the survival rate of patients [170,216,224]. Thus, it has been proposed that at least in some tumors the expression level of the β1i and β5i subunits might represent a predictive marker of response to ICKi [216,225]. As a matter of fact, the overexpression of these subunits is linked to longer survival and improved response to the ICKi therapy in melanoma patients and the proposed mechanism underlying this connection consists in enhanced reactivity of TILs toward melanoma cells as a consequence of an altered repertoire of the presented antigens [216]. Importantly, it has also been reported that immunoproteasome subunit overexpression sometimes occurs regardless of IFNγ and T cell infiltration, suggesting that these subunits should be independent prognostic biomarker with respect to the IFNγ level and the rate of T cell infiltration in the context of tumor microenvironment [216]. Thus, this last observation opens up a possible and yet poorly investigated scenario concerning the role of immunoproteasome in mediating the ICKi response independently of the IFNγ pathway. Therefore, even though many crucial points deserve to be clarified, it clearly emerges that immunoproteasome expression seems to be an important predictive marker in colorectal cancer, melanoma, and NSCLC, for which the ICKi therapy has proven to be effective. Thus, an intriguing therapeutic strategy that should be explored in the near future is the combination of ICKi and drugs that directly modulate immunoproteasome activity and/or induce immunoproteasome expression in order to increase its pool inside the cells.

## 6. Conclusions

Cancer cells are often more dependent on a proper integrity and functionality of UPS than nonmalignant cells due to the rapid proliferation rate, increased metabolic activity, and continuous exposure to a variety of extrinsic stress perturbations (such as nutrient deprivation, hypoxia, and acidosis) under which cancer cells live. All these conditions lead to a decrease in protein quality control and make UPS a suitable target for cancer therapy [218,219]. Accordingly, a number of proteasome based-strategies have been proposed, spanning from (i) inhibition of proteasome proteolytic activity, (ii) modulation of the abundance of proteasome regulatory particles (i.e., 19S or PA28) and of their interaction with the 20S to (iii) modulation of the activity of enzymes involved in proteasome subunit post-translational modifications and (iv) interference with transport of natural low-molecular-weight proteasome activators (e.g., spermine) [62,226,227,228,229,230,231,232]. Additionally, a series of strategies targeting the ubiquitination cascade have been studied. One of the most intriguing and novel therapeutic approaches involves the use of PROTAC (proteolysis-targeting chimeric molecules), which are hetero-bifunctional molecules that recruit specific target proteins to the E3 ligase, thus inducing the increase of target ubiquitination and degradation. This strategy has already been applied to the degradation of a number of selected targets [233,234,235]. However, even though promising, it is still in its infancy for application to immunotherapy. On the other hand, the most deeply investigated strategy consists in the use of broad-specificity inhibitors of the 20S activity, like bortezomib, carfilzomib, and ixazomib that inhibit proteolysis of all the proteasome forms present in different cells [15,62]. The second approach encompasses the identification of specific inhibitors targeting inducible tissue-specific forms of proteasome, mainly the i20S, which is involved in the production of antigenic peptides. Though this mechanism is well-known, how such inhibitors might either decrease or stimulate cancer cell recognition by T cells is debated (see Section 4) [62]. As a matter of fact, a challenging but poorly investigated issue is the precise in vivo impact of broad-specificity 20S proteasome inhibitors commonly used in clinical practice on antigen presentation by cancer cells [99], and it should deserve more attention. A key question for improving the efficacy and safety profile of immunotherapy includes the identification of the most appropriate strategy to optimize the antigenic peptide repertoire of the tumor required for an efficient immune response [99]. Notably, some recent data suggest that enhanced immunoproteasome activity might play an important role in the response of melanoma to ICKi [216]. It seems to indicate that, at least in some tumors, the more efficient strategy could be to “enhance” immunoproteasome expression and activity. Thus, the choice of the best strategy whether to inhibit or activate immunoproteasome should take into consideration the biological features of the specific tumor that has to be treated. Even though additional in vitro and in vivo investigation needs to be performed, the current evidence indicates that selective modulation of proteasome activity might have a role in improving the outcome of the ICKi or other immunotherapeutic approaches.

## Figures and Tables

**Figure 1 cancers-13-04852-f001:**
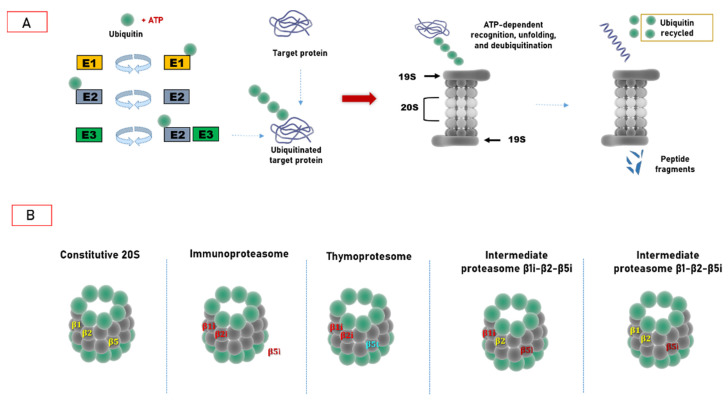
Ubiquitin–proteasome system. (**A**) General UPS organization. The Ub target protein conjugation cascade consists of three classes of ubiquitin ligases: E1, E2, and E3. Ub is activated by E1, forming a high-energy thioester bond; then, activated Ub is transferred to E2, and then E3 attaches Ub to a specific polypeptide substrate. At the end stage, the substrates are recognized and subjected to ATP-dependent degradation by the 26S proteasome. (**B**) Structural heterogeneity of the proteasome core particle; 20S constitutive proteasome contains the standard catalytic subunits β1 (caspase-like activity), β2 (trypsin-like activity), and β5 (chymotrypsin-like activity). The immunoproteasome incorporates the inducible catalytic subunits β1i, β2i, and β5i, which have different catalytic specificities. The thymoproteasome is expressed only by cortical thymic epithelial cells and is characterized by the presence of β1i-β2i, and β5t subunits. Finally, intermediate proteasomes contain a mix of constitutive and inducible subunits: β1i,-β2-β5i and β1-β2, and β5i, respectively.

**Figure 2 cancers-13-04852-f002:**
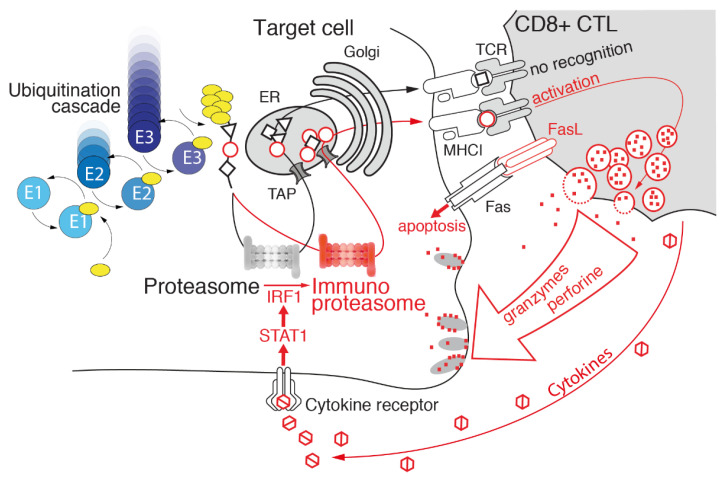
Immunoproteasome mediates cell cytotoxicity. The development of an inflammatory reaction, particularly the production of IFNγ and other inflammatory cytokines, induces the expression of the proteasome immune subunits in target cells and leads to efficient production of peptides beneficial in terms of presentation on the MHC class I and recognizable by CD8+ CTLs. Further elevation of the immunoproteasome (in red) mediated by increased production of inflammatory cytokines and subsequent generation of immunodominant peptides may function as a positive feedback loop, enhancing T cell-mediated cytotoxicity.

**Figure 3 cancers-13-04852-f003:**
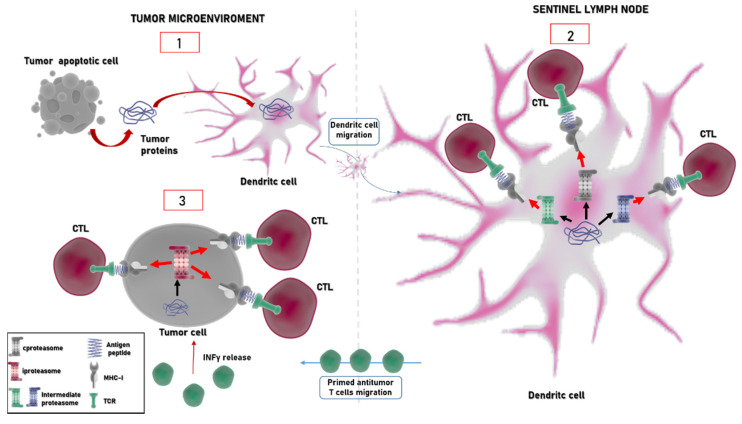
Activation of antitumor T cells mediated by proteasomes. At the tumor site, apoptotic cells release tumor proteins which are internalized by dendritic cells (1). These cells migrate to the sentinel lymph node, where peptides are derived by proteasome (mainly constitutive (in grey) and intermediate proteasome (in green and in blue) particles). Different populations of proteasome are represented with different colors in the figure, as indicated also in the legend. Degradation of tumor-associated proteins is presented to naïve T cells in complex with MHC-I molecules (2). Primed T cells migrate to the tumor site (3). The in loco production of IFNγ induces the expression of i26S (in red) which becomes the prevalent form of proteasome in the “hot tumor site”. Thus, tumor cells with a high content of immunoproteasome change their antigenic peptide repertoire, which can be targeted by other circulating T cells.

**Figure 4 cancers-13-04852-f004:**
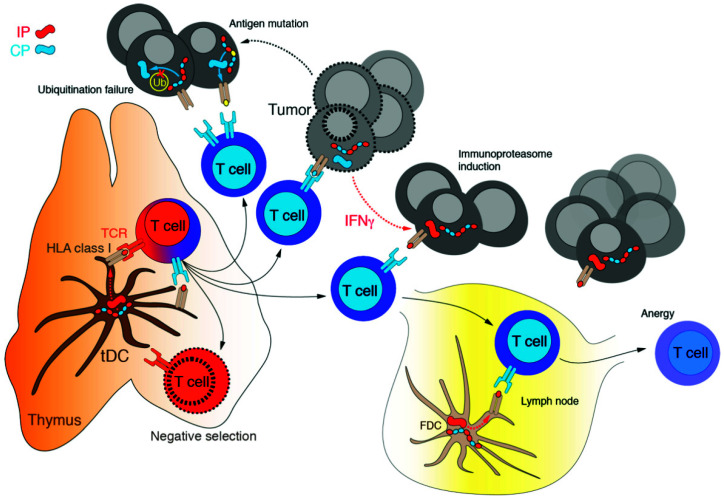
Mechanisms of tumor immunoescape. T cell clones that passed negative selection recognize a limited number of tumor self-antigens. Moreover, cancer cells may escape from cytotoxic CD8+ T cells by (i) IFNγ-mediated induction of immunoproteasome with altered pattern of antigens processing, (ii) mutation of the antigen leading to the failure of the recognition by TCR, and (iii) inactivation of intracellular ubiquitination of tumor antigens caused by either suppression of expression or functional mutation of the E2/E3 ligases specific to tumor antigens.

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
