# Peer review of "At the Cutting Edge against Cancer: A Perspective on Immunoproteasome and Immune Checkpoints Modulation as a Potential Therapeutic Intervention"

_cancers, 2021, doi:10.3390/cancers13194852_

Round 1

Reviewer 1 Report

The manuscript by Tundo et al. reviews the knowledge about (immuno)proteasomes in cancer immunotherapy and the crosstalk between proteasome modulators and immune checkpoint inhibitors. Herein, the authors focus on the role of the ubiquitin proteasome system (UPS)and proteasome isoforms in antigen presentation. They start with an introduction of cancer immunotherapy in general, followed by general descriptions of the UPS, proteasome isoforms including proteasome inhibitors, and finish their descriptions with a chapter on immunoproteasomes and check point inhibitors. Overall, these parts are written well and presented nicely. However, there is a bit of overinterpretation on the role of immunoproteasomes in antigen presentation and some concepts are missing. At some points a bit more focus on the topic would be good. Also the choice of references does not always refer to more recent reviews or new original publications.

  1. The review would benefit from shortening all parts, which are not directly correlated to the major topic: immunoproteasome and immune-check points (i.e. extensive descriptions of different types of E3 ligases, Ub-linkages, gating, thymoproteasomes).
  2. Paragraph 3.2. on immunoproteasomes can be kept short and its function in antigen presentation and beyond can be combined with the chapter on immunoproteasomes and cancer.
  3. The DRIP hypothesis and their impact in antigen presentation is completely missing and should be explained by a few sentences.
  4. Using at least 3 different icons for proteasomes in different figures is puzzling. The same icons for proteasomes (isoforms in different colors) and substrates should be used throughout the different figures to help the reader in understanding of these complex cellular pathways.
  5. Fig 3: In the text the authors correctly explain that there are immunoproteasome inducers other than IFN-g (IFN-a and b, TNF-a etc.). This is also valid in a tumor microenvironment where myeloid cells or fibroblasts produce these alternative cytokines. Indicate this fact for instance by “cytokine receptor” instead of IFNg and IFNgR. The figure should be modified here accordingly. There is a typo for “Golgi” in the target cell.
  6. Fig 4: the 3 different colors for proteasomes are not explained in legend. It is misleading to show 26S proteasomes in tumor cells or DCs and describe i20S for immunoproteasomes in the legend and throughout the paper. Generation of antigenic peptides is mainly ubiquitin-dependent and 26S proteasomes are required. Please, change to i26S proteasomes as abbreviation.
  7. Lines 387-403: It is clearly described that the contribution of proteasome isoforms to antigen presentation is strictly dependent on the cell type and the epitope; immunoproteasomes even destroy tumor epitopes in some cases. This should be clarified. The authors should tone down their exclusive description of immunoproteasomes as peptide producers in context of a tumor to give a balanced view. Moreover, also for KO mice for single immunosubunits the contribution to MHC class I antigen presentation ranges from 10 to almost 50% in splenic cells, however ß5i contribution is similar to that of triple KO (see doi: 10.1126/science.8066463; doi: 1038/ni.2203). Nevertheless, KO mice are viable and still have 50% MHC class I antigen presentation left. That means that standard proteasomes considerably contribute to general MHC class I antigen presentation, but also to specific epitopes. This fact is also underlined by the well-known fact that general inhibition of the proteasome blocks generation of peptides presented on MHC class I molecules. Here, the authors should consider also the contribution of the UPS in general and cite more recent reviews on antigen presentation (for example: Special issue “The Immunoproteasome in Health and Disease” in Cells (MDPI).
  8. Chapter 4 on immunoproteasome: an emerging cancer target:

As the authors mentioned above immunoproteasomes have important functions beyond antigen processing including in regulation of many pathways (NFkB and others). This should be considered in the discussion of immunoproteasome-specific inhibitors or proteasome inhibitors in cancer in general. More importantly, (immuno)proteasome function in several immune pathways (see also reviews on UPS in immune cells/ signaling) as well proteasomal (or lysosomal) degradation of immune check-point molecules such PD1L (i.e. naturally or promoted by a PROTAC) should be discussed carefully also in view of therapeutical interventions.

Minor:

Fig. 2 right side: correct peptides!

beta5t is PSMB11 and not PSMB17

Box 2: A focus on tumor antigens is highly appreciated here to stay in the scope of the paper.

Author Response

Reviewer 1

The manuscript by Tundo et al. reviews the knowledge about (immuno)proteasomes in cancer immunotherapy and the crosstalk between proteasome modulators and immune checkpoint inhibitors. Herein, the authors focus on the role of the ubiquitin proteasome system (UPS)and proteasome isoforms in antigen presentation. They start with an introduction of cancer immunotherapy in general, followed by general descriptions of the UPS, proteasome isoforms including proteasome inhibitors, and finish their descriptions with a chapter on immunoproteasomes and check point inhibitors. Overall, these parts are written well and presented nicely. However, there is a bit of overinterpretation on the role of immunoproteasomes in antigen presentation and some concepts are missing. At some points a bit more focus on the topic would be good. Also the choice of references does not always refer to more recent reviews or new original publications.

Reviewer: The review would benefit from shortening all parts, which are not directly correlated to the major topic: immunoproteasome and immune-check points (i.e. extensive descriptions of different types of E3 ligases, Ub-linkages, gating, thymoproteasomes).

Paragraph 3.2. on immunoproteasomes can be kept short and its function in antigen presentation and beyond can be combined with the chapter on immunoproteasomes and cancer

Answer: The reviewer's suggestion is correct. Thus, some parts of the manuscript have been kept short and other parts have been edited and re-organized. Anyway, in our opinion, the 3 and 4 chapters should remain divided. Obviously, if reviewer deems it necessary to render the paper suitable for the publication, we will combine the two chapters.

Reviewer: The DRIP hypothesis and their impact in antigen presentation is completely missing and should be explained by a few sentences.

Answer: We acknowledge the reviewer’s suggestion. In the revised version, the DRIP hypothesis has been introduced and discussed.

Reviewer: Using at least 3 different icons for proteasomes in different figures is puzzling. The same icons for proteasomes (isoforms in different colors) and substrates should be used throughout the different figures to help the reader in understanding of these complex cellular pathways.

Answer: The figures have been edited according to the reviewer’s suggestions without changing the style of the figures.

Reviewer: Fig 3: In the text the authors correctly explain that there are immunoproteasome inducers other than IFN-g (IFN-a and b, TNF-a etc.). This is also valid in a tumor microenvironment where myeloid cells or fibroblasts produce these alternative cytokines. Indicate this fact for instance by “cytokine receptor” instead of IFNg and IFNgR. The figure should be modified here accordingly. There is a typo for “Golgi” in the target cell.

Answer: The figure has been edited according to reviewer’s suggestions.

Reviewer: Fig 4: the 3 different colors for proteasomes are not explained in legend. It is misleading to show 26S proteasomes in tumor cells or DCs and describe i20S for immunoproteasomes in the legend and throughout the paper. Generation of antigenic peptides is mainly ubiquitin-dependent and 26S proteasomes are required. Please, change to i26S proteasomes as abbreviation.

Answer: Non-stimulated cells constitutively express a major amount of constitutive and intermediate proteasome, whereas immunoproteasome is the predominant population inside cytokine-stimulated cells. 26S is formed by 20S and 19S; 20S can incorporate both standard and inducible subunits and, therefore, it can be constitutive (c20S) or inducible (i20S). c20S and i20S can bind 19S to form c26S and i26S, respectively. Probably, the mistake in the legend was misleading. In the revised manuscript, we have corrected the figure legend.

Reviewer: Lines 387-403: It is clearly described that the contribution of proteasome isoforms to antigen presentation is strictly dependent on the cell type and the epitope; immunoproteasomes even destroy tumor epitopes in some cases. This should be clarified. The authors should tone down their exclusive description of immunoproteasomes as peptide producers in context of a tumor to give a balanced view. Moreover, also for KO mice for single immunosubunits the contribution to MHC class I antigen presentation ranges from 10 to almost 50% in splenic cells, however ß5i contribution is similar to that of triple KO (see doi: 10.1126/science.8066463; doi: 1038/ni.2203). Nevertheless, KO mice are viable and still have 50% MHC class I antigen presentation left. That means that standard proteasomes considerably contribute to general MHC class I antigen presentation, but also to specific epitopes. This fact is also underlined by the well-known fact that general inhibition of the proteasome blocks generation of peptides presented on MHC class I molecules. Here, the authors should consider also the contribution of the UPS in general and cite more recent reviews on antigen presentation (for example: Special issue “The Immunoproteasome in Health and Disease” in Cells (MDPI).

Answer: We would like to thank the reviewer for the suggestion. The specific part of the main text has been modified accordingly.

Reviewer: Chapter 4 on immunoproteasome: an emerging cancer target:

As the authors mentioned above immunoproteasomes have important functions beyond antigen processing including in regulation of many pathways (NFkB and others). This should be considered in the discussion of immunoproteasome-specific inhibitors or proteasome inhibitors in cancer in general. More importantly, (immuno)proteasome function in several immune pathways (see also reviews on UPS in immune cells/ signaling) as well proteasomal (or lysosomal) degradation of immune check-point molecules such PD1L (i.e. naturally or promoted by a PROTAC) should be discussed carefully also in view of therapeutical interventions.

Answer: According to the reviewer’s suggestion, the specific part of the main text has been modified in sections 3, 4 and 5.

Answer: The mistakes have been edited

Reviewer: Box 2: A focus on tumor antigens is highly appreciated here to stay in the scope of the paper.

Answer: In the revised version of manuscript, BOX 2 has been deleted.

Reviewer 2 Report

    In this review article, the authors attempted to discuss the potential relationship between immunoproteasome and immune checkpoint inhibitors. Although this was a potentially interesting topic, as described in the title and abstract, the vast majority of the main text was about unrelated, or slightly related topics. Only a small portion of the article was for the crosstalk between immunoproteasome and immune checkpoints. I’m willing to support this article if the major concerns are fully addressed, and unrelated topics are deleted or summarized.

Major concerns:

  1. The vast majority of the main text was about unrelated, or slightly related topics. The authors spent the majority of the efforts on the “background” information, such as ubiquitin, constitutive proteasome and thymoproteasome. It’s difficult for a reader, like me, to stay focus on unrelated information. Because the title and abstract were written as such, please focus on immunoproteasome, immune checkpoints, and the crosstalk between them. Please summarize or delete the rest of the background information.
  2. There was very little evidence to support that immunoproteasome had any relationship with immune checkpoints. One observation provided by authors was the weak association between beta1i/beta5i and clinical responses. One alternative explanation was that the increase of beta1i/beta5i simply reflected the increase of immune cell infiltration, but not the up-regulation of beta1i/beta5i in tumor cells. Please focus on identifying evidence that agrees or disagrees the hypothesis.
  3. It appears to me that the only potential mechanism between checkpoint inhibitors and immunoproteasome are the IFN-gamma signaling pathway. However, it’s only briefly mentioned in the section 5 (Line 659). Please provide more evidence and expand the topic significantly. Please also identify other potential links between checkpoint inhibitors and immunoproteasome. The current evidence presented in the text was very little.
  4. One of the major functions of the immunoproteasome is antigen presentation. The authors only spent a little effort on the topic of antigen presentation and the changes of immune epitopes. In Line 551-554, the authors mentioned that some of the MAGE epitopes were generated by “intermediate” proteosome. Please focus on this topic, and discuss how immunoproteasome may generate different epitopes, especially the neoantigen epitopes. Otherwise, the evidence to support the hypothesis that immunoproteasome is important for immunotherapy is very weak.
  5. The ratio between constitutive 20S proteasome, intermediate proteasome and immunoproteasome was likely transient and flexible, not an all-or-none event (Line 341-344). Therefore, the antigens processed by 20S or immunoproteasome may co-exist in tumor cells, just the ratio could be different depending on the tumor microenvironment. Please address this concern.
  6. Section 5.1 was focused on immunoproteasome “inhibitors”. Based on the hypothesis in this article, my understanding is that we should look for “enhancers”, not inhibitors. Please explain.

Minor concerns:

  1. The abbreviation for interferon-gamma is IFN-gamma, not INF-gamma. INF was used 12 times, and IFN was used 8 times.
  2. I don’t believe the BOX part (Line 705-833) is required by the Cancers. The text is too long for a box. Please delete or summarize them.
  3. Please include “Simple Summary”. I believe it’s required by Cancers.

Author Response

Reviewer 2

In this review article, the authors attempted to discuss the potential relationship between immunoproteasome and immune checkpoint inhibitors. Although this was a potentially interesting topic, as described in the title and abstract, the vast majority of the main text was about unrelated, or slightly related topics. Only a small portion of the article was for the crosstalk between immunoproteasome and immune checkpoints. I’m willing to support this article if the major concerns are fully addressed, and unrelated topics are deleted or summarized.

Reviewer: The vast majority of the main text was about unrelated, or slightly related topics. The authors spent the majority of the efforts on the “background” information, such as ubiquitin, constitutive proteasome and :  thymoproteasome. It’s difficult for a reader, like me, to stay focus on unrelated information. Because the title and abstract were written as such, please focus on immunoproteasome, immune checkpoints, and the crosstalk between them. Please summarize or delete the rest of the background information.

Answer: We acknowledge the criticism raised by the reviewer. Probably, the title and abstract were misleading and, therefore, in the revised version of the manuscript they have been modified. The scope of the review is to provide an overall picture of the role of immunoproteasome in antigen presentation to support the hypothesis about novel therapeutic interventions to be combined with immune checkpoint inhibitors. Unfortunately, the studies on the cross-talk between immunoproteasome and immune checkpoint inhibitors are rare. In our opinion, this aspect deserves more attention and we hope that this review might contribute to raise the interest on this topic.

Reviewer: There was very little evidence to support that immunoproteasome had any relationship with immune checkpoints. One observation provided by authors was the weak association between beta1i/beta5i and clinical responses. One alternative explanation was that the increase of beta1i/beta5i simply reflected the increase of immune cell infiltration, but not the up-regulation of beta1i/beta5i in tumor cells. Please focus on identifying evidence that agrees or disagrees the hypothesis.

Answer: Yes, it is correct that until now, little evidence supporting association between immunoproteasome and immune checkpoint inhibitors are reported. However, it is not due to a real lack of association, but it simply reflects the low number of available studies right now. Therefore, according to the reviewer’s suggestions, we have modified the title and the abstract to better clarify the topic of the manuscript. Overexpression of immunoproteasome subunits has been reported to be predictive of better survival and improved response to immune-checkpoint inhibitors of melanoma patients. The mechanism proposed underlying this connection is that overexpression induces an alteration of tumor associated peptide presented by TILs (see DOI: 10.1016/j.bioorg.2021.104833). We have added a sentence in the section 5 to better clarify this concept.

Reviewer: It appears to me that the only potential mechanism between checkpoint inhibitors and immunoproteasome are the IFN-gamma signaling pathway. However, it’s only briefly mentioned in the section 5 (Line 659). Please provide more evidence and expand the topic significantly. Please also identify other potential links between checkpoint inhibitors and immunoproteasome. The current evidence presented in the text was very little.

Answer: The topic has been expanded as indicated by reviewer.

Reviewer: One of the major functions of the immunoproteasome is antigen presentation. The authors only spent a little effort on the topic of antigen presentation and the changes of immune epitopes. In Line 551-554, the authors mentioned that some of the MAGE epitopes were generated by “intermediate” proteosome. Please focus on this topic, and discuss how immunoproteasome may generate different epitopes, especially the neoantigen epitopes. Otherwise, the evidence to support the hypothesis that immunoproteasome is important for immunotherapy is very weak.

Answer: We fully agree with the reviewer’s comments. The identification of neoantigens in general and how antigens are differentially processed by the immune and constitutive proteasome is a promising area of research, due to its clinical relevance. Therefore, multidisciplinary approaches have to be continuously developed to identify these novel targets.

Reviewer: The ratio between constitutive 20S proteasome, intermediate proteasome and immunoproteasome was likely transient and flexible, not an all-or-none event (Line 341-344). Therefore, the antigens processed by 20S or immunoproteasome may co-exist in tumor cells, just the ratio could be different depending on the tumor microenvironment. Please address this concern.

Answer: We fully agree with the reviewer’s comments, as reported in different points of the main text. In the revised manuscript, the concept has been also reiterated where suggested by the reviewer.

Reviewer: Section 5.1 was focused on immunoproteasome “inhibitors”. Based on the hypothesis in this article, my understanding is that we should look for “enhancers”, not inhibitors. Please explain.

Answer: The Section 4.1 (not 5.1) is focused on immunoproteasome inhibitors because one of the main topic of the research on immunoproteasome is the discovery of drugs selectively targeting this form of proteasome, in order to reduce the several drawbacks associated with constitutive proteasome inhibitors of clinical interest (i.e., low potency and specificity, onset of adverse effects and development of drug resistance),. As the reviewer says and as reported in the “conclusion” section, an alternative strategy, at least in some tumours, should be the “identification” of “enhancers”. The correct therapeutic strategy is probably related to the biological features of the specific tumour that has to be treated. As already mentioned, the simple scope of this review is to contribute to raise this scientific problem. The conclusion section has been edited to better explain this point.

Minor concerns:

Reviewer: The abbreviation for interferon-gamma is IFN-gamma, not INF-gamma. INF was used 12 times, and IFN was used 8 times.

Answer: Thanks for the suggestion. We have edited the text.

Reviewer: I don’t believe the BOX part (Line 705-833) is required by the Cancers. The text is too long for a box. Please delete or summarize them.

Answer: Box 2 has been deleted

Reviewer: Please include “Simple Summary”. I believe it’s required by Cancers.

Answer: In the revised version of the manuscript, a simple summary has been included.

Reviewer 3 Report

The authors reviewed the cross-talk of immunoproteasome and ICKi. The topic is interesting and review is comprehensive and well written. Few suggestions to help improve:

  1. Can authors also provide the idea how the host microenvironment contributes to the TME?
  2. The existing markers such as TMB, PD-L1, etc, are facing some challenges and negative results, such as a recent paper (1016/j.annonc.2021.02.006). Can authors provide their insights on this?
  3. In clinical translational side, it is always preferable to have a non-invasive measure of TME. As such, imaging-based radiogenomics analysis has been proposed, see our latest review (1016/j.semcancer.2020.12.005). Also, in our recent study, we found imaging subtype is predictive to ICKi in NSCLC (https://www.nature.com/articles/s42256-021-00377-0). I’d like to hear the authors comment how can imaging contribute to help decode the cross-talk.

Author Response

Reviewer 3

The authors reviewed the cross-talk of immunoproteasome and ICKi. The topic is interesting and review is comprehensive and well written. Few suggestions to help improve:

Reviewer : Can authors also provide the idea how the host microenvironment contributes to the TME?

Answer: In the revised manuscript, a sentence regarding the host microenvironment contribute to the tumour microenvironment and its influence on the efficacy or immune checkpoint inhibitors has been added.

Reviewer: The existing markers such as TMB, PD-L1, etc, are facing some challenges and negative results, such as a recent paper (1016/j.annonc.2021.02.006). Can authors provide their insights on this?

Answer: Thanks for the suggestion. We have edited the text accordingly.

Reviewer: In clinical translational side, it is always preferable to have a non-invasive measure of TME. As such, imaging-based radiogenomics analysis has been proposed, see our latest review (1016/j.semcancer.2020.12.005). Also, in our recent study, we found imaging subtype is predictive to ICKi in NSCLC (https://www.nature.com/articles/s42256-021-00377-0). I’d like to hear the authors comment how can imaging contribute to help decode the cross-talk.

Answer: We have edited the text according to the referee’s suggestions.

Round 2

Reviewer 2 Report

For the Simple Summary, please read the instruction and use plain language.